# FoxO1–Dio2 signaling axis governs cardiomyocyte thyroid hormone metabolism and hypertrophic growth

Anwarul Ferdous [1], Zhao V. Wang [1], Yuxuan Luo[1], Dan L. Li[1], Xiang Luo [1], Gabriele G. Schiattarella [1], Francisco Altamirano [1], Herman I. May [1], Pavan K. Battiprolu[1], Annie Nguyen [1], Beverly A. Rothermel[1,2], Sergio Lavandero [1,3], Thomas G. Gillette [1] & Joseph A. Hill [1,2✉]

Forkhead box O (FoxO) proteins and thyroid hormone (TH) have well established roles in cardiovascular morphogenesis and remodeling. However, specific role(s) of individual FoxO family members in stress-induced growth and remodeling of cardiomyocytes remains unknown. Here, we report that FoxO1, but not FoxO3, activity is essential for reciprocal regulation of types II and III iodothyronine deiodinases (Dio2 and Dio3, respectively), key enzymes involved in intracellular TH metabolism. We further show that *Dio2* is a direct transcriptional target of FoxO1, and the FoxO1–Dio2 axis governs TH-induced hypertrophic growth of neonatal cardiomyocytes in vitro and in vivo. Utilizing transverse aortic constriction as a model of hemodynamic stress in wild-type and cardiomyocyte-restricted *FoxO1* knockout mice, we unveil an essential role for the FoxO1–Dio2 axis in afterload-induced pathological cardiac remodeling and activation of TRα1. These findings demonstrate a previously unrecognized FoxO1–Dio2 signaling axis in stress-induced cardiomyocyte growth and remodeling and intracellular TH homeostasis.

[1] Department of Internal Medicine (Cardiology Division), University of Texas Southwestern Medical Center, Dallas, TX 75390-8573, USA. [2] Department of Molecular Biology, University of Texas Southwestern Medical Center, Dallas, TX 75390-8573, USA. [3] Advanced Center for Chronic Diseases (ACCDiS) and Corporacion Centro de Estudios Cientificos de las Enfermedades Cronicas (CECEC), Universidad de Chile, Santiago 8380492, Chile. ✉email: joseph.hill@utsouthwestern.edu

Heart failure remains a leading cause of death in industrialized nations, and the epidemic is rapidly expanding to include the developing world[1]. A variety of disease-related stimuli, such as myocardial infarction, ischemia, and chronic hypertension, trigger pathological cardiac remodeling, a process that often progresses to heart failure and sudden death[2–4]. Conventional thinking holds that hypertrophic growth of the myocardium is a compensatory response to increases in workload, serving to minimize wall stress and preserve contractile function. However, accumulating evidence, both preclinical and epidemiological, highlights maladaptive features of chronic ventricular hypertrophy[5–7], suggesting that targeting of cardiac hypertrophy may be a relevant therapeutic goal[8]. Numerous transcriptional and signaling pathways have been identified in the regulation of pathological cardiac remodeling[9]. Among them, the Forkhead box O (FoxO) family of transcription factors has emerged as critical drivers of stress-induced cardiac remodeling[10].

FoxO factors are a subclass of the large family of Forkhead transcriptional regulators characterized by a conserved 110 amino acid DNA-binding motif called the "forkhead box" or "winged helix" domain[11]. The forkhead domain of four members of the FoxO subgroup (FoxO1, FoxO3, FoxO4, and FoxO6) recognizes a consensus DNA-binding element, 5′-RYAAAYA-3′ (where R = A/G, Y = C/T), in numerous genes; indeed, the fact that this FoxO-responsive element (FRE) is recognized by all FoxO factors suggests that aspects of their functions overlap[12]. However, targeted disruption of FoxO1, but not FoxO3 or FoxO4, in mice causes embryonic lethality due to abnormal cardiovascular and placental morphogenesis[13–16]. Moreover, we have recently reported that FoxO1, but not FoxO3, activity is critically associated with metabolic stress-induced pathological remodeling of the heart[17]. These observations support a specific role of FoxO1 in cardiovascular morphogenesis and metabolic stress-induced cardiac remodeling that cannot be compensated by other FoxO family members[18]. On the other hand, combined inactivation of FoxO1 and FoxO3 in adult cardiomyocytes has been shown to exacerbate ischemic damage to the myocardium[19], whereas mice lacking FoxO4 are resistant to ischemic damage to the heart[20]. Moreover, mice lacking FoxO3 are sensitized to transverse aortic constriction (TAC)-induced cardiac hypertrophy[21,22]. Collectively, these studies demonstrate an essential but distinct role of FoxO factors in cardiac remodeling and that the nature of external stimuli differentially impacts the activity of each FoxO factor. However, molecular mechanisms underlying FoxO1 action in stress-induced hypertrophic remodeling of cardiomyocytes remain largely unknown.

A growing literature points to post-translational modifications, such as phosphorylation, acetylation, and ubiquitination, as predominant mechanisms that regulate FoxO activity[12,23,24]. It is now well established that phosphorylation of FoxO factors by Akt following activation of insulin or insulin-like growth factor-1 (IGF-1) receptors negatively regulates FoxO activity, stability, and subcellular localization[11]. More recently, thyroid hormones (THs) have been reported to potentiate FoxO1 activity in hepatocytes by inhibiting Akt activity[25], thereby unfolding another layer of complexity in the orchestrated control of FoxO activity. The physiological significance of such a FoxO1–TH signaling axis in cardiomyocyte heath has yet to be elucidated.

TH has long been implicated in cardiomyocyte health in the developing, neonatal, and adult heart[26]. In humans, abnormal TH levels in the fetus and neonate are linked to multiple cardiovascular complications, including diminished cardiac output and tachycardia[27]. Importantly, subtle changes in TH homeostasis are also intimately linked with cardiovascular disease[28,29], highlighting the fact that THs are critical regulators of cellular homeostasis in most tissues[30,31]. Although circulating levels of the prohormone 3,5,3′,5′-tetraiodothyronine (thyroxine or T4) and the active isoform 3,5,3′-L-triiodothyronine (T3) are commonly measured clinically to evaluate an individual's thyroid status, less well recognized is the fact that THs are metabolized intracellularly. Specifically, much of TH action in muscle cells is directly regulated by two important deiodinase enzymes: the type II iodothyronine deiodinase (Dio2) is involved in active TH biosynthesis by converting the inactive prohormone T4 to active isoform T3, and the type III deiodinase (Dio3) inactivates both T4 and T3 (refs. [31,32]).

In light of the established roles of both FoxO1 and TH in disease-related cardiac remodeling, coupled with the interplay between them in some settings, we set out to address two major questions: (a) Does a FoxO–Dio2 signaling axis contribute to stress-induced hypertrophic remodeling of cardiomyocytes? (b) Does FoxO activity govern deiodinase gene expression in cardiomyocytes to regulate TH metabolism? Here, we demonstrate that FoxO1 activity is essential for reciprocal regulation of Dio2 and Dio3 expression and that the FoxO1–Dio2 signaling axis governs TH- and stress-induced cardiomyocyte hypertrophic growth and pathological remodeling of the heart.

## Results

**FoxO1 governs TH-induced cardiomyocyte growth by inversely regulating *Dio2* and *Dio3* expression.** To gain insight into the role of FoxO factors in TH-induced cardiomyocyte growth, we treated neonatal rat ventricular myocytes (NRVMs) in culture with control and two sequence-independent *FoxO1*-specific small interfering RNAs (siRNAs) and then assessed for TH-induced NRVM growth using a leucine incorporation assay. We noted that robust, T4-induced growth of control siRNA-treated cells was attenuated in FoxO1-deficient cells (Fig. 1a, Supplementary Fig. 1A). Efficient knockdown of *FoxO1* mRNA (Fig. 1b) and protein (Fig. 1c, d) levels was confirmed using quantitative RT-PCR (qPCR) and immunoblot analyses, respectively.

Next, we sought to determine whether FoxO1 activity plays a role in cardiomyocyte growth in response to other stimuli. To test this, we assessed NRVM growth after treating the cells with a variety of physiological (IGF-1) and pathological [phenylephrine (PE) and angiotensin II (Ang II)] hypertrophic agonists. We noted that unlike T4-induced cardiomyocyte growth, knockdown of FoxO1 had little impact on the growth response elicited by other agonists (Fig. 1a). Importantly, NRVM growth in response to other agonists (e.g. Ang II) in FoxO1-deficient cells trended toward an increase relative to control siRNA-treated cells (Fig. 1a). These observations support a specific role of FoxO1 in TH-induced cardiomyocyte growth and are concordant with our previous study demonstrating that FoxO1 gain-of-function inhibits Ang II-induced NRVM growth[21].

In addition to T4, knockdown of *FoxO1* also resulted in significant attenuation of T3-induced NRVM growth (Supplementary Fig. 1A). These data suggest that FoxO1-dependent regulation of intracellular TH metabolism is required for TH-induced NRVM growth. We, therefore, analyzed mRNA levels of deiodinase genes associated with TH metabolism and observed that selective knockdown of *FoxO1* significantly attenuated both *Dio2* mRNA (Fig. 1b) and protein (Fig. 1c, d) levels. By contrast, knockdown of *FoxO1* resulted in marked induction of *Dio3* expression, whereas *Dio1* expression remained unchanged (Fig. 1b). Based on these findings, we reasoned that induction of *Dio3* in FoxO1-deficient cells may inactivate T3 action. To test this, we treated cells with control or two sequence-independent *Dio3*-specific siRNAs and confirmed efficient knockdown of *Dio3* mRNA levels (Supplementary Fig. 1B). Next, NRVMs were treated with control, *FoxO1*- or *Dio3*-specific siRNAs alone or in

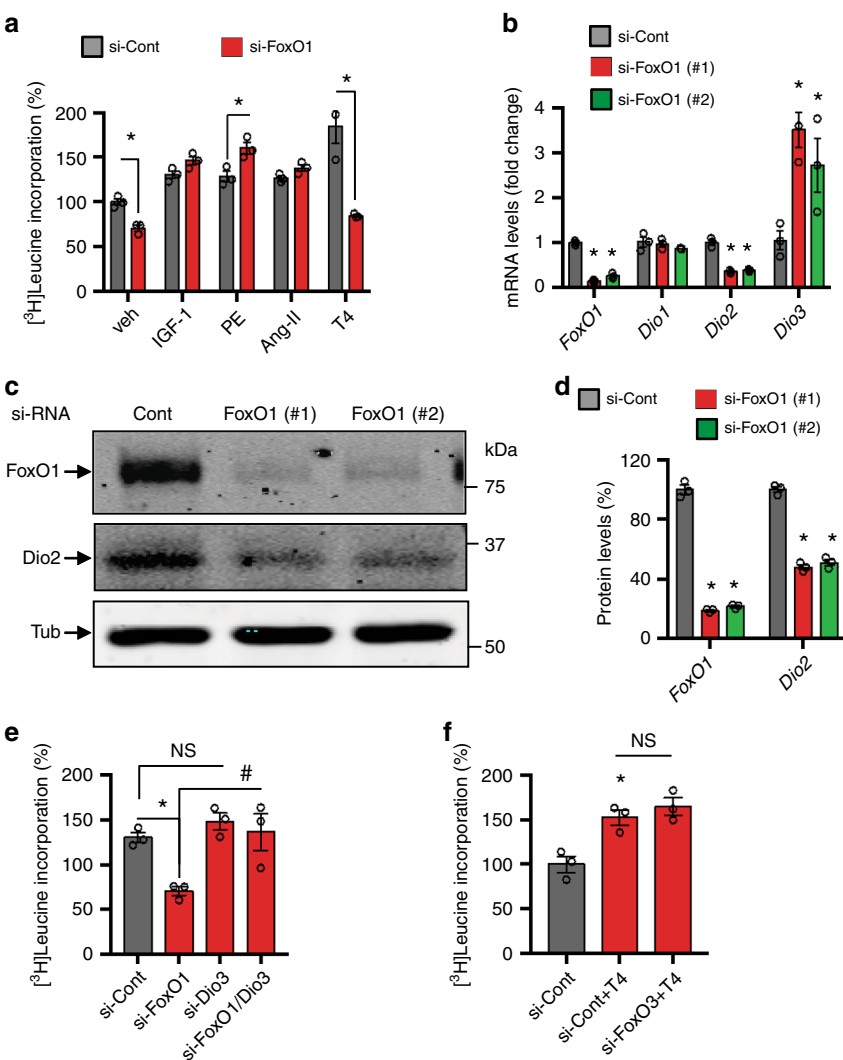

**Fig. 1 FoxO1 and Dio2/Dio3 transcriptional circuitry govern TH-induced NRVM growth in vitro. a** Selective knockdown of *FoxO1* in NRVM specifically abrogated T4-induced hypertrophy but not the cellular growth response triggered by other stimuli. NRVM growth was evaluated by assessing radiolabeled leucine incorporation into protein following 24 h treatment, where NRVM growth in the control (Cont) siRNA- and vehicle (Veh)-treated cells was set to 100%. **b** Selective knockdown of *FoxO1* in NRVM resulted in marked reduction of *Dio2* mRNA levels and significantly induced *Dio3* expression. **c**, **d** Immunoblotting (**c**) and quantitation (**d**) of FoxO1 and Dio2 levels in FoxO1-deficient NRVM. **e** T3-induced growth response of NRVM transfected with control, *Dio3*- and *FoxO1*-specific siRNAs alone or in combination. Note that the abrogation of T3-induced NRVM growth response in FoxO1-deficient cells was largely rescued in cells transfected with both *Dio3*- and *FoxO1*-specific siRNAs. **f** Selective knockdown of *FoxO3* did not affect T4-induced growth response of NRVM. In all panels, data are depicted as mean ± SEM ($n = 3$ independent experiments). $*p < 0.05$ vs control; $\#p < 0.01$ vs control, NS not statistically significant. Statistical analyses were conducted using a two-tailed, unpaired Student's $t$-test.

combination, and then assessed T3-induced NRVM growth. We observed that abrogation of the T3-induced NRVM growth response in FoxO1-deficient cells was largely rescued by concomitant knockdown of *Dio3*, and that T3-induced NRVM growth was not inhibited in Dio3-deficient cells (Fig. 1e), suggesting that TH metabolism by deiodinase enzymes plays an important role in TH-induced cell growth.

It has been reported that FoxO3, a member of the FoxO family with actions functionally distinct from FoxO1, governs *Dio2* expression in skeletal muscle cells[33]. To test whether FoxO3 activity also governs *Dio2* expression in cardiomyocytes, we depleted cells of FoxO3 using two sequence-independent *FoxO3*-specific siRNAs and noted that efficient knockdown of *FoxO3* did not alter *Dio2* or *Dio3* mRNA levels (Supplementary Fig. 1C); as expected, however, mRNA levels of two known FoxO3 targets were significantly attenuated[34] (Supplementary Fig. 1D). These findings suggest that FoxO3 is not the primary regulator of *Dio2*

in cardiomyocytes. Consistent with this notion, selective knockdown of *FoxO3* did not alter the T4-triggered growth response of NRVM (Fig. 1f). We conclude that FoxO1, but not FoxO3, activity inversely regulates *Dio2* and *Dio3* expression in cardiomyocytes and that FoxO1–Dio2/Dio3 transcriptional circuitry governs TH-induced cardiomyocyte growth.

### FoxO1–Dio2 axis governs TH-induced hypertrophic growth of cardiomyocytes. To test further for a specific role of the FoxO1–Dio2 axis in TH-induced cardiomyocyte remodeling, we treated NRVM with control or two sequence-independent *Dio1*- and *Dio2*-specific siRNAs. Although efficient knockdown of *Dio1* mRNA levels did not affect T4-induced NRVM growth (Supplementary Fig. 2A, B), selective knockdown of *Dio2* mRNA (Fig. 2a) and protein (Supplementary Fig. 2C) blunted T4-induced NRVM growth (Fig. 2b). In addition, lack of change in

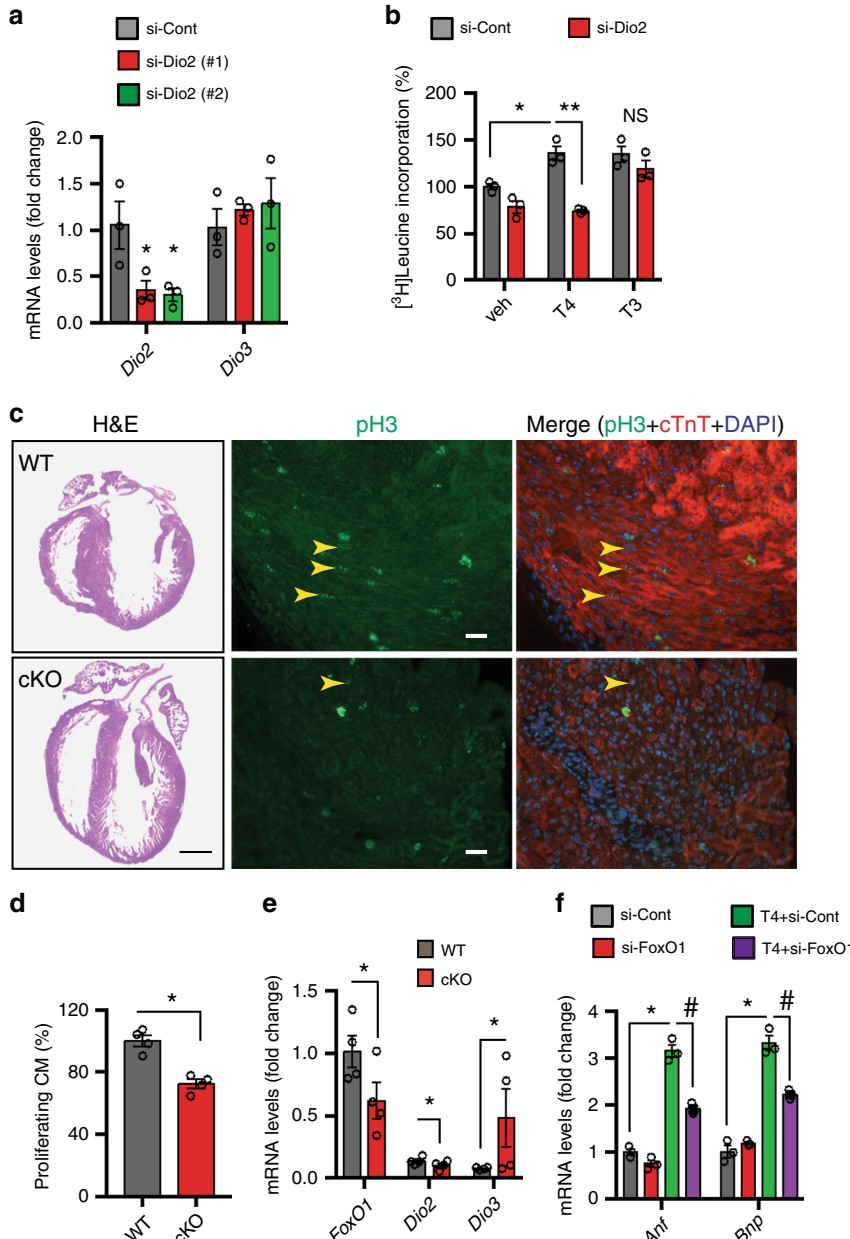

**Fig. 2 FoxO1–Dio2 axis governs TH-induced hypertrophic growth of neonatal cardiomyocytes. a** qRT-PCR analyses of mRNA levels of the indicated genes in NRVM transfected with control or two sequence-independent *Dio2*-specific siRNAs. Note that efficient knockdown of *Dio2* did not affect *Dio3* expression (*n* = 3 independent experiments). **b** Selective knockdown of *Dio2* in NRVM abrogated the T4-induced, but not T3-induced, cellular growth response (*n* = 3 independent experiments). **c** Histology (left) (bar = 2 mm) and co-immunostaining with anti-pH3 (green) (middle) and anti-cTnT (red) (right) antibodies of P3 *FoxO1-WT* (WT) and *FoxO1-cKO* (cKO) heart. Proliferating cardiomyocytes in LV free wall are shown (arrowhead), and DAPI was used for the nuclear staining (bar = 10 μm). **d** Quantitation of proliferating cardiomyocytes in WT and cKO hearts (*n* = 4), where total number of proliferating cells in WT hearts was considered to be 100%. Note the significant decrease in proliferating cells in cKO heart. **e** qRT-PCR analyses of ventricular mRNA levels of the indicated genes in P3 WT and cKO mice, and mRNA levels of ribosomal 18S was used as control (*n* = 4). **f** qRT-PCR analyses of mRNA levels of the indicated genes in control- and si-FoxO1-transfected NRVM with or without T4 treatment (*n* = 3 independent experiments). In all panels, data are depicted as mean ± SEM. *$p < 0.05$ vs control; **$p < 0.01$ vs control; #$p < 0.001$ vs control. Statistical analyses were conducted using a two-tailed, unpaired Student's *t*-test.

*Dio3* mRNA levels in Dio2-deficient cells (Fig. 2a) implies that Dio2 activity is not required for T3-induced cell growth. Indeed, selective knockdown of *Dio2* did not affect T3-induced cell growth (Fig. 2b). Together, these data further support the notion that the FoxO1–Dio2 axis plays an important role in TH-induced cardiomyocyte growth.

Next, we tested the role of the FoxO1–Dio2 axis in cardiomyocyte growth in vivo. Neonatal mouse cardiomyocytes

manifest robust proliferative capacity for the first few days of life[35]. Compared with hypoxic conditions of embryonic heart, neonatal heart growth takes place in a highly oxidative environment, and that growth is associated with profound changes in cardiac contractility in response to increased metabolic and hemodynamic demand[36,37]. As FoxOs and TH are important regulators of metabolic and redox events[38], we sought to determine whether FoxO1 activity participates in

neonatal cardiomyocyte growth. We harvested postnatal day 3 (P3) hearts from wild-type (WT) mice and performed immuno-histochemical analyses of a mitotic marker (phospho-histone H3, pH3) and a cardiomyocyte-specific marker (cardiac troponin T). Dual-positive nuclei were counted as proliferating cardiomyo-cytes[35]. Using this technique, we detected large numbers of proliferating cardiomyocytes in left ventricle (LV) free wall (Fig. 2c, Supplementary Fig. 2D).

To evaluate FoxO1-specific regulation of cardiomyocyte proliferation, we crossed mice harboring floxed alleles of *FoxO1* (*FoxO1^{L/L}*)[15] (Supplementary Fig. 2E) with αMHC-Cre, a cardiomyocyte-specific Cre recombinase-expressing mouse line, to allow for cardiomyocyte-restricted inactivation of *FoxO1*. The resulting *FoxO1^{L/L}/αMHC-Cre* and *FoxO1^{L/L}* mice are referred to herein as cKO and WT, respectively. Cardiomyocyte-specific inactivation of *FoxO1* was confirmed using semi-quantitative PCR comparing Cre-mediated recombination of floxed sites in the genomic DNA (gDNA) isolated from heart (ventricles) and tail of WT and cKO mice (Supplementary Fig. 2E, F) as well as qPCR analyses of ventricular *FoxO1* mRNA levels in WT and cKO hearts (Fig. 2e). We noted that compared with WT hearts, the number of proliferating cardiomyocytes was significantly reduced in cKO mice (Fig. 2c, d), coinciding with reciprocal regulation of *Dio2* and *Dio3* expression in WT and cKO hearts (Fig. 2e). These findings support a role for the FoxO1–Dio2/Dio3 axis in redox-induced cardiomyocyte remodeling.

We further noted that compared with vehicle-treated cells, T4 treatment of control siRNA-treated cells induced transcript levels of two hypertrophic markers, *Anf* (*Nppa*) and *BNP* (*Nppb*), in NRVM[27], whereas knockdown of *FoxO1* markedly attenuated T4-induced hypertrophic marker expression (Fig. 2f). Together, we conclude that the FoxO1–Dio2 signaling axis participates in stress-induced hypertrophic growth of cardiomyocytes.

**FoxO1 activity in early post-TAC heart reciprocally regulates *Dio2* and *Dio3* expression.** To test whether the FoxO1–Dio2 axis also participates in maladaptive cardiomyocyte remodeling in the setting of disease-related stress, we examined expression of FoxO1 target genes in LV of WT mice exposed to TAC surgery, a pressure overload-induced stress model of hyper-tension causing robust cardiac hypertrophy[39,40]. Total RNA was extracted from LVs 4 days post-TAC as well as from control (sham)-operated mice, and mRNA levels of selected FoxO1 target genes were evaluated by qPCR. Compared with sham-operated hearts, we noted robust up-regulation of mRNA levels of several known FoxO1 targets in TAC LV[16,41,42] (Fig. 3a). By contrast, mRNA levels of *CamkIIδ* and a FoxO3 target, *Murf1* (ref.[34]), were not induced in TAC-stressed LV (Fig. 3a). These data suggest that activation of select FoxO1 targets in early TAC-stressed LV is not a result of global acti-vation of gene expression. Immunoblot analyses to monitor FoxO1 phosphorylation status at Thr24 and Thr32 revealed significant decreases in phosphorylated (inactive) FoxO1 levels in TAC-stressed LV protein lysates (Fig. 3b). Together, these data reveal that pressure overload triggers activation of FoxO1 in early afterload-stressed heart.

We also evaluated expression of the so called "fetal gene program", a collection of genes reactivated by pathological stress. Compared with sham-operated hearts, we observed robust induction of transcript levels of several hypertrophic markers, including *Myh7* (i.e. β-MHC), *Anf* (*Nppa*), *BNP* (*Nppb*), and *Rcan1.4* (also called MCIP1), a proxy for calcineurin/NFAT activity, in TAC hearts, along with a concomitant decrease in *Myh6* (i.e. α-MHC) message (Supplementary Fig. 3A). These data indicate that in addition to FoxO1, the hypertrophic gene

program and signaling cascade are activated early in load-stressed LV.

To validate FoxO1-specific activation of its target genes in cardiomyocytes of stressed heart, we analyzed mRNA levels of FoxO1 targets in cKO hearts. Cardiomyocyte-specific inactivation of *FoxO1* was confirmed by analyzing Cre-mediated recombina-tion of floxed sites in genomic DNA of isolated ventricular cardiomyocytes (Supplementary Fig. 3B) and by immunoblotting of LV lysates (Supplementary Fig. 3C), respectively. We also noted that cardiomyocyte-restricted silencing of FoxO1 did not alter FoxO3 protein levels (Supplementary Fig. 3C). Importantly, cardiomyocyte-specific inactivation of FoxO1 blunted induction of its target genes in TAC-stressed LV (Supplementary Fig. 3D), corroborating FoxO1-specific activation of target genes in afterload-stressed heart.

Compared with sham-operated heart, we observed significant increases in *Dio2* mRNA levels in TAC-stressed hearts of both WT and cKO mice; however, induction of *Dio2* expression was significantly attenuated in cKO heart (Fig. 3c). By contrast, mRNA levels of *Dio1* (Supplementary Fig. 3E) and *Dio3* (Fig. 3d) were not induced in TAC-stressed hearts of WT mice, whereas mRNA levels of *Dio3*, but not *Dio1*, in cKO LV were higher at baseline than in WT littermates and markedly induced in post-TAC LV (Fig. 3d). Collectively, these data lend strong credence to our hypothesis that FoxO1 is a primary regulator of reciprocal expression of *Dio2* and *Dio3* in cardiomyocytes. We also hypothesized that FoxO1 is a direct, upstream regulator of the *Dio2* gene.

Sequence analysis revealed two conserved *F*oxO-responsive *e*lements (FREs) in the upstream promoter region of the *Dio2* gene[33] (Supplementary Fig. 3F). Utilizing quantitative analyses of chromatin immunoprecipitation (ChIP) assays in sham- and TAC-operated ventricles, we noted significant increases in FoxO1 occupancy at the *Dio2* promoter in TAC-stressed hearts compared with sham or control IgG (Fig. 3e). These data suggest that FoxO1 binding at the conserved FREs in the upstream *Dio2* promoter is required for robust activation of *Dio2* expression in stressed hearts.

To define the specificity of FoxO1 as a transcriptional regulator of *Dio2*, we fused the *Dio2* promoter to a luciferase reporter and conducted transcriptional assays. Co-transfection in COS7 cells with a *Dio2* reporter construct harboring WT-FREs with increasing amounts of caFoxO1 expression plasmid resulted in marked and concentration-dependent induction of luciferase activity (Fig. 3f). By contrast, mutation of the FREs (Supplemen-tary Fig. 3G) blunted transcriptional activity (Fig. 3f). Collectively, these data suggest that FoxO1 binding to the FREs is essential for its transcriptional activity and robust induction of *Dio2* expression in TAC-stressed WT hearts.

**FoxO1–Dio2 axis governs TAC-induced cardiac hypertrophy and contractile dysfunction.** To determine the functional sig-nificance of the FoxO1–Dio2 signaling axis in stress-induced pathological cardiac hypertrophy, normal chow-fed WT mice were subjected to sham or TAC surgery. The fetal gene program, hypertrophic growth, and cardiac function and fibrosis were evaluated 3 weeks post-TAC surgery, a time point when hyper-trophic growth reaches steady state[43]. We noted robust cardiac hypertrophy, contractile dysfunction, fibrosis and activation of fibrotic and fetal gene programs in TAC-operated mice when compared with sham-operated mice (Fig. 4, Supplementary Fig. 4). By contrast, this hypertrophic phenotype was markedly attenuated in TAC-operated WT mice fed propylthiouracil (PTU, a potent inhibitor of deiodinase enzymes)-containing chow for 1 week before being subjected to surgery (Supplementary Fig. 4).

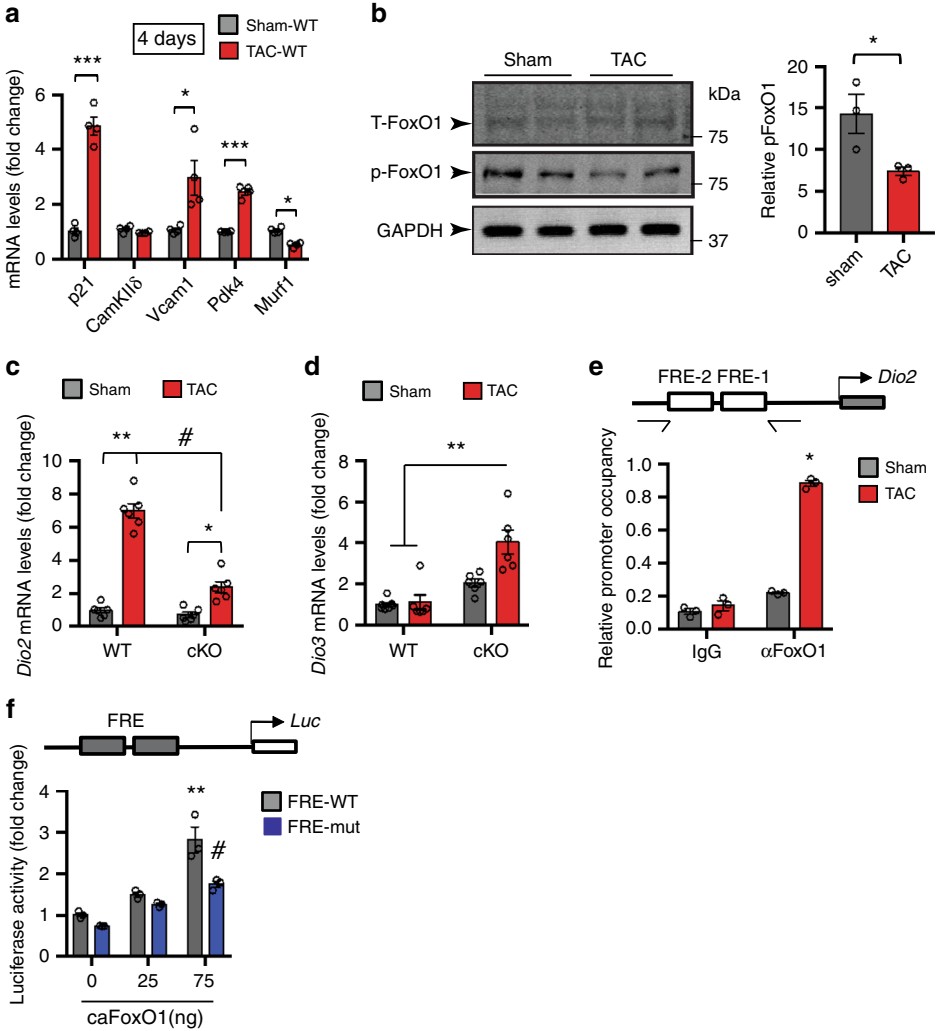

**Fig. 3 Activation of FoxO1 in early post-TAC heart is required for reciprocal regulation of *Dio2* and *Dio3* genes expression. a** qRT-PCR analyses of mRNA levels of indicated genes in 4 day TAC- and sham-operated LVs ($n = 4$). **b** Immunoblotting (left) and quantitation (right) of total-FoxO1 (T-FoxO1) and phospho-FoxO1 (p-FoxO1) levels in 4 days TAC- and sham-operated LVs ($n = 3$). GAPDH was used as a loading control. Average of p-FoxO1/T-FoxO1 and p-FoxO1/GAPDH ratios were used to determine relative p-FoxO1 levels. **c, d** qRT-PCR analyses of mRNA levels of *Dio2* (**c**) and *Dio3* (**d**) in 4 days TAC- and sham-operated LVs of *FoxO1-WT* (WT, $n = 12$) and *FoxO1-cKO* (cKO, $n = 12$) mice. **e** Quantitative ChIP assays were conducted in 4 day sham- and TAC-operated ventricles of the WT mice ($n = 3$). Note the significant increase in FoxO1 occupancy at the *Dio2* promoter in TAC hearts. Schematic of *Dio2* illustrates that PCR amplification was performed using primers spanning the two FoxO-responsive elements (FREs) in the *Dio2* promoter. **f** Schematic of *Dio2* luciferase vector harboring two FREs. Note that co-transfection of a constitutively active FoxO1 (caFoxO1) elicited significant, concentration-dependent activation of reporter activity harboring WT, but not mutated, FREs. In all panels, data are depicted as mean ± SEM. *$p < 0.05$ vs sham/control; **$p < 0.01$ vs sham; ***$p < 0.0001$ vs sham, #$p < 0.05$ vs WT (**f**); #$p < 0.001$ vs control (**c**). Statistical analyses were conducted using a two-tailed, unpaired Student's $t$-test.

Consistent with our previous report[44], these data support the notion that attenuation of cardiac hypertrophy in PTU chow-fed mice was due, in part, to blockade of Dio2-mediated T3 biosynthesis, consistent with the role of Dio2, but not Dio1, in TH-induced NRVM growth (Fig. 2a, Supplementary Fig. 2B) and robust induction of *Dio2* expression in early TAC-stressed hearts (Fig. 3c).

To test this, we quantified mRNA levels of *Dio1* and *Dio2* genes in normal chow-fed WT LVs and noted that relative *Dio1* expression is significantly less than that of *Dio2* (Supplementary Fig. 5A). By contrast, semi-quantitative RT-PCR analyses revealed more *Dio1* transcripts than *Dio2* in liver (Supplementary Fig. 5B), implying a tissue-specific differential expression of *Dio1* and *Dio2* genes. Therefore, we also analyzed *Dio1* and *Dio2* expression in ventricular cardiomyocytes isolated from sham- and TAC-operated hearts and noted robust induction of *Dio2*

(Supplementary Fig. 5C) and a hypertrophic marker *βMHC* (Supplementary Fig. 5D) expression in TAC-stressed cardiomyocytes. Importantly, compared with *Dio2*, *Dio1* expression in isolated cardiomyocytes was essentially undetectable (Supplementary Fig. 5C). Moreover, only T4-induced cell growth was markedly attenuated when we rendered NRVMs "intracellularly hypothyroid" by exposing them to PTU (Supplementary Fig. 5E). On the other hand, selective knockdown of *Dio2*, but not *Dio1*, resulted in T4-induced NRVM growth insensitive to PTU treatment (Supplementary Fig. 5F). Collectively, these data further support our hypothesis that the FoxO1–Dio2 axis plays an essential role in cardiomyocyte TH homeostasis and stress-induced pathological cardiac hypertrophy.

Pituitary Dio2 activity is critically linked with metabolic stress-induced adipocyte hypertrophy and obesity[45]. To exclude a global effect of PTU on Dio2 as well as on Dio1 activity, we assessed

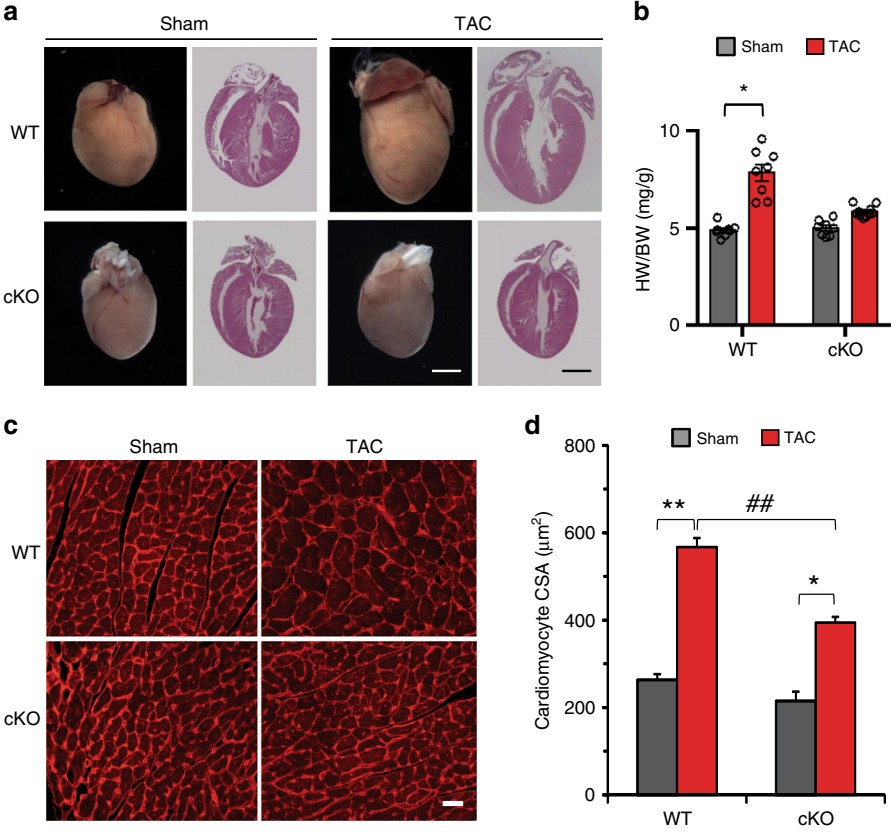

**Fig. 4 FoxO1 is required for pathological cardiomyocyte growth. a** Male mice of respective genotypes, *FoxO1-WT* (WT) and *FoxO1-cKO* (cKO), were subjected to sham and TAC surgery. Whole mount (left) and histology (right) of representative 3-week post-sham and -TAC hearts are shown. Note that cardiomyocyte-specific silencing of *FoxO1* (cKO) blunted TAC-induced hypertrophic growth of heart (bar = 2 mm). **b** Heart weight (HW)/body weight (BW) ratios indicate robust cardiac hypertrophy in WT ($n = 16$) mice 3-week post-TAC, which is attenuated in cKO mice ($n = 18$). *$p < 0.001$ vs sham. **c**, **d** Immunohistochemistry (**c**) and quantitation (**d**) of cross-sectional area (CSA) of cardiomyocytes of WT and cKO mice after 3-week sham and TAC surgery ($n = 3$). Data are depicted as mean ± SEM. *$p < 0.05$ vs sham; **$p < 0.01$ vs sham; ##$p < 0.01$ vs WT. Statistical analyses were conducted using a two-tailed, unpaired Student's *t*-test.

TAC-induced cardiac hypertrophy in 8- to 10-week-old cKO, control littermates (WT), and age-matched *αMHC*-Cre-expressing control mice (herein termed Cre). Mice of each genotype with normal cardiac function, as evaluated by transthoracic echocardiography, were exposed to TAC surgery along with a cohort exposed to sham surgery. Echocardiography was performed 3 weeks post-surgery, and body weight was measured prior to sacrificing the animals to harvest hearts or LVs for further analysis. Compared with sham-operated hearts, gross analysis and histology of TAC-stressed hearts of WT mice manifested clear evidence of robust hypertrophic growth (Fig. 4a). Hypertrophic growth of post-TAC hearts was also observed in Cre mice (Supplementary Fig. 6A). By contrast, TAC-induced cardiac hypertrophy was significantly blunted in cKO mice (Fig. 4a). These observations were corroborated by measurements of heart weight/body weight ratios in which control hearts manifested marked increases (nearly 60%) in size, whereas cKO mice displayed only modest (~16%) growth (Fig. 4b). To confirm that this cardiac growth was due to hypertrophic growth of cardiomyocytes, cross-sectional area (CSA) of ≈100 cardiomyocytes was measured in transverse sections of LV septa stained with wheat germ agglutinin (Fig. 4c). Compared with sham-operated WT mice, we observed a modest decrease in heart and cardiomyocyte size in cKO mice (Fig. 4a, d). However, cardiomyocytes of both WT and cKO mice manifested significant increases in CSA after TAC surgery, although that hypertrophic growth of cardiomyocytes in cKO hearts was attenuated

relative to WT littermates (Fig. 4d). Collectively, our in vitro and in vivo data suggest that in response to upstream stimuli, the FoxO1–Dio2 axis acts as an essential driver of cardiomyocyte hypertrophic growth.

To assess whether TAC-induced increases in FoxO1 activity are required to promote maladaptive cardiac hypertrophy, we evaluated cardiac function in conscious mice by transthoracic echocardiography. Three weeks after TAC surgery, contractile performance quantified as left ventricular fractional shortening (FS) was decreased in WT (Fig. 5a, b) and Cre (Supplementary Fig. 6B) mice compared with sham-operated littermates. Interestingly, the decline in FS after TAC in cKO mice was significantly less than that in WT mice (Fig. 5a, b). The reduction in FS after TAC in WT mice was driven by increases in both LV end-diastolic diameter (LVEDD) and LV end-systolic diameter (LVESD) (Fig. 5c, d). Importantly, cKO mice did not exhibit significant change in LVEDD and LVESD compared to sham (Fig. 5c, d). Together, these data suggest that activation of FoxO1 in cardiomyocytes early after TAC is essential to drive pressure overload-induced maladaptive cardiac remodeling and contractile dysfunction.

**Ablation of cardiomyocyte *FoxO1* attenuates TAC-induced fetal gene reactivation and cardiac fibrosis.** We evaluated whether attenuation of hypertrophic growth in cKO hearts is associated with reduced expression of hypertrophic markers.

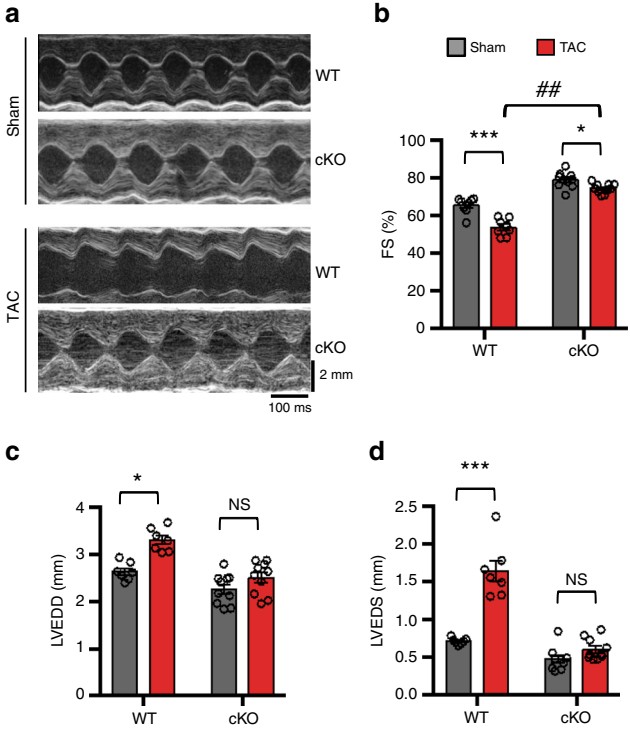

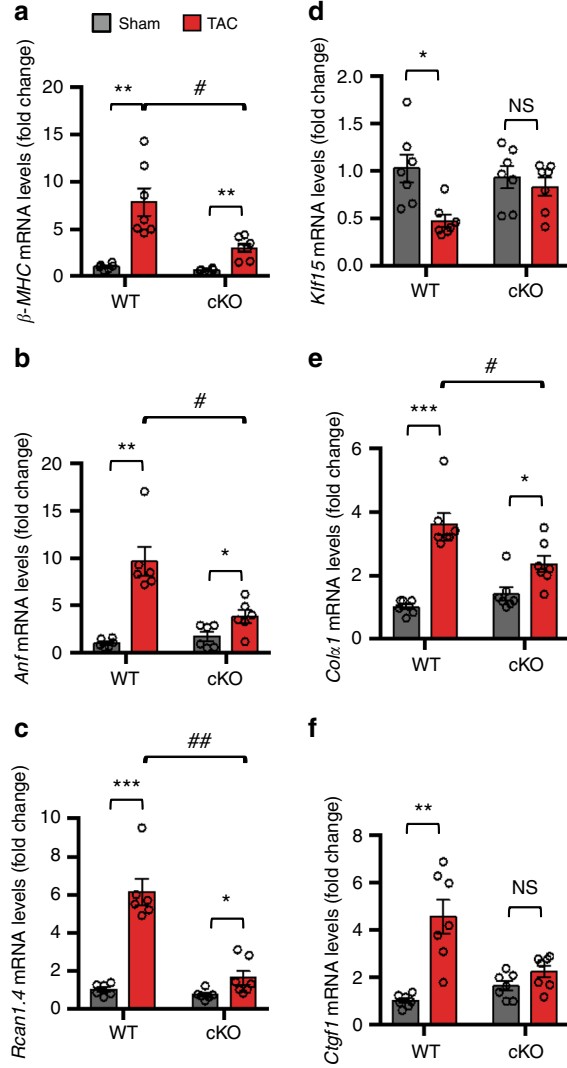

**Fig. 5 FoxO1-cKO hearts manifest preserved contractile function in response to TAC. a** Representative M-mode tracings of 3-week sham- and TAC-operated hearts of *FoxO1-WT* (WT) and *FoxO1-cKO* (cKO) mice. **b** Percent of left ventricular fractional shortening (%FS) of WT (*n* = 16) and cKO (*n* = 20). **c** Left ventricular end-diastolic diameter (LVEDD, mm) of WT (*n* = 14) and cKO (*n* = 20), and **d** Left ventricular end-systolic diameter (LVESD, mm) of WT (*n* = 14) and cKO (*n* = 19) are shown. Note that cKO mice do not manifest contractile dysfunction when compared with WT littermate TAC heart. In all panels, data are depicted as mean ± SEM. *p < 0.01 vs sham; ***p < 0.001 vs sham; ##p < 0.0001 vs cKO; NS not statistically significant. Statistical analyses were conducted using a two-tailed, unpaired Student's *t*-test.

Indeed, we noted robust induction of transcript levels of hypertrophic markers, including *β-MHC*, *Rcan1.4*, and *Nppa* in Cre (Supplementary Fig. 6C) and in WT (Fig. 6a–c) TAC hearts, along with a concomitant decrease in *α-MHC* message (Supplementary Fig. 7A). By contrast, induction of these genes was markedly attenuated in cKO TAC heart (Fig. 6a–c). Further, in contrast with the robust induction of hypertrophic markers in TAC-stressed WT hearts, expression of *Klf15*, a transcriptional inhibitor of pathological cardiac hypertrophy[46], was significantly attenuated in TAC-stressed WT, but not in cKO, hearts (Fig. 6d).

Excessive fibrogenesis due to activation of cardiac fibroblasts in the setting of disease-related stress contributes to cardiac contractile and electrical dysfunction and ultimately heart failure[47]. Compared with sham-operated hearts, trichrome staining revealed marked increases in reactive interstitial fibrosis in TAC-stressed hearts of Cre (Supplementary Fig. 6D) and WT mice (Supplementary Fig. 7B). In contrast, interstitial fibrosis was essentially absent in TAC-stressed cKO hearts (Supplementary Fig. 7B). The increases in cardiac fibrosis coincided with significant induction of transcript levels of fibrotic genes, including *Colα1*, *Ctgf1*, and *α-SMA* in TAC-stressed WT LV, but not cKO, mice (Fig. 6e, f, Supplementary Fig. 7C). These data support the notion that mice lacking *FoxO1* in cardiomyocytes are less susceptible to pressure overload-induced maladaptive cardiac remodeling, hypertrophy-associated gene expression, and fibrosis.

**Fig. 6 Reactivation of fetal and fibrotic gene programs in response to TAC is blunted in cKO hearts. a–f** qRT-PCR analyses of mRNA levels of the indicated genes in 3-week TAC and sham LVs of *FoxO1-WT* [WT, *n* = 13 (**a**), 12 (**b, c**), and 14 (**d–f**)] and *FoxO1-cKO* [cKO, *n* = 13 (**a**), 12 (**b, c**), and 14 (**d–f**)] mice. mRNA levels of ribosomal 18S were used as control. In all panels, data are depicted as mean ± SEM. *p < 0.05 vs Sham; **p < 0.01 vs sham; ***p < 0.001 vs sham; #p < 0.05 vs WT, ##p < 0.001 vs WT; NS not statistically significant. Statistical analyses were conducted using a two-tailed, unpaired Student's *t*-test.

**FoxO1-dependent induction of *Dio2* governs intracellular TH action via TH-responsive gene expression.** TH triggers cellular growth by activating mitogen activated protein kinases (MAPK, e.g. ERK1/2)[48], transcriptional activity of the thyroid hormone receptor (THR), the FoxO1–Akt signaling axis, and autophagy[26,28,29]. Importantly, pathological cardiac remodeling has been reported to be associated with induced *Dio2* expression[49], p38-dependent repression of *Klf15* expression[50], and increased autophagic activity[51] in hearts. In light of this, and having detected reduced *Klf15* expression (Fig. 6d) and increased autophagic activity[51] in WT TAC hearts, we tested whether ERK activity is also increased in load-stressed hypertrophic hearts. Immunoblotting of LV lysates revealed increased ERK activity in WT TAC hearts, a change not seen in cKO TAC hearts (Supplementary Fig. 8A). Importantly, increased *Dio2* expression in TAC-stressed WT LV (Fig. 3c) coincided with marked induction

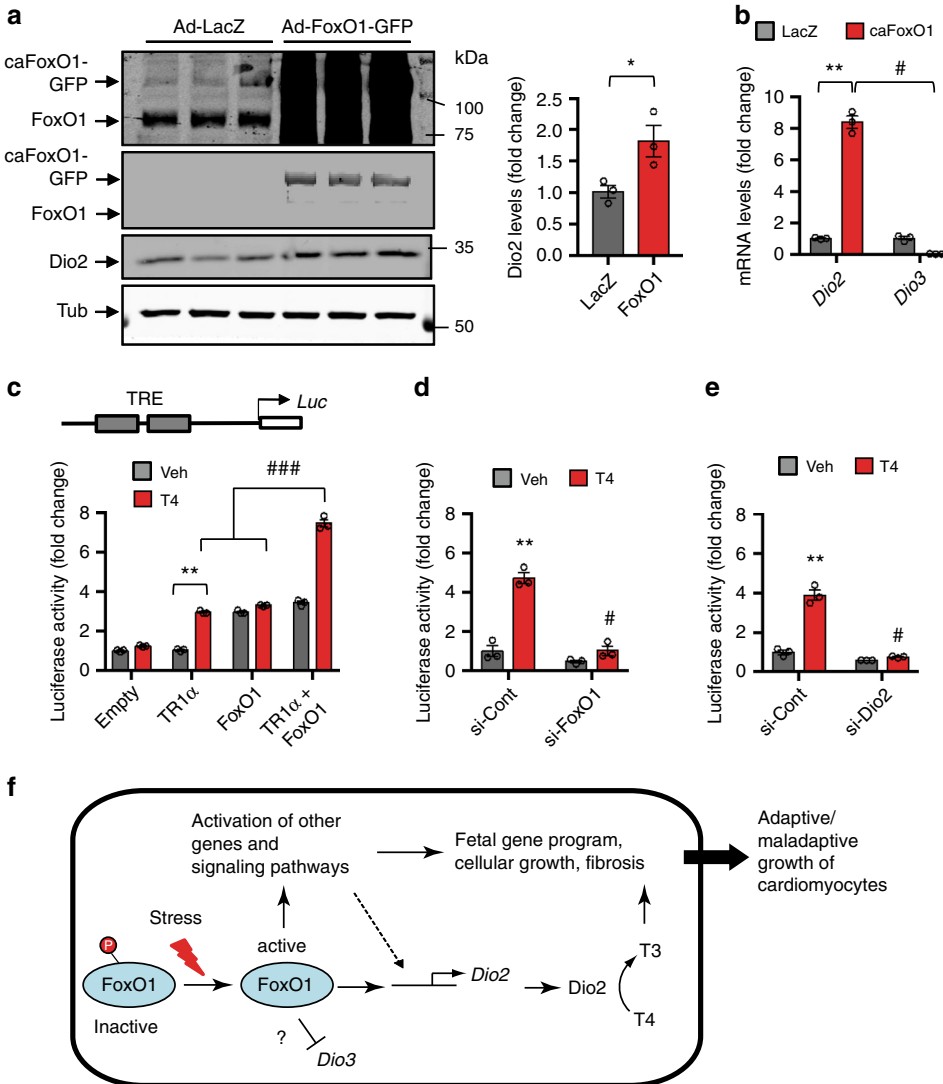

**Fig. 7 FoxO1-dependent induction of *Dio2* governs TH-responsive gene expression in hypertrophic cardiomyocytes. a**, **b** caFoxO1 overexpression in NRVM markedly induced Dio2 protein (**a**) and mRNA (**b**) levels but repressed *Dio3* gene expression (**b**). **c** Schematic of thyroid hormone receptor (THR)-responsive reporter plasmid harboring two thyroid hormone-responsive elements (TREs). Note that co-expression of caFoxO1 and THRα1 (TRα1), but not either alone, in HEK293 cells resulted in robust and synergistic activation of reporter activity only in the presence of T4 (*n* = 3). **d**, **e** THR-responsive reporter activity in NRVM transfected with control or indicated gene-specific (*FoxO1* and *Dio2*) siRNAs. Note that compared with control siRNA-treated cells, selective knockdown of *FoxO1* (**d**) and *Dio2* (**e**) significantly attenuated T4-dependent activation of luciferase activity. In all panels, data are depicted as mean ± SEM (*n* = 3 independent experiments). *$p < 0.05$ vs control; **$p < 0.001$ vs control; #$p < 0.001$ vs control; ###$p < 0.0001$ vs control. Statistical analyses were conducted using a two-tailed, unpaired Student's *t*-test. **f** Working model for FoxO1's role in adaptive and maladaptive hypertrophic growth of cardiomyocytes. In response to oxidative- and TAC-induced (Stress), activation of FoxO1 and FoxO1-dependent reciprocal regulation of *Dio2* and *Dio3* in cardiomyocytes is a key driver for subsequent activation of other transcriptional and signaling programs to activate fetal and fibrotic genes, which together ultimately lead to adaptive and maladaptive hypertrophic growth of neonatal and adult cardiomyocytes, respectively.

of mRNA levels of *Mag*, an established THR downstream target[52], only in WT TAC hearts (Supplementary Fig. 8B).

These data suggest that activation of THR is associated with FoxO1–Dio2 axis-mediated increases in local active TH levels. However, we found that quantification of T3 or T4 levels in sham- and TAC-operated LV lysates was technically challenging, possibly due to the extremely short half-life of T3 in rodents[53] as well as T3/T4 contamination from blood and non-cardiomyocyte cells in LV tissue. This may explain why cardiomyocyte-specific overexpression of *Dio2* in mice yielded modest increases in cardiac T3 levels without changing cardiac or circulating T4 levels[54].

To test whether FoxO1-dependent induction of Dio2 is linked to T3-dependent activation of THR, we first employed a FoxO1 gain-of-function strategy in NRVMs. We found that compared with control (LacZ) protein, adenovirus-mediated overexpression of caFoxO1 in NRVMs resulted in marked increases in both Dio2 protein (Fig. 7a) and mRNA (Fig. 7b). By contrast, overexpression of FoxO1 significantly attenuated *Dio3* expression (Fig. 7b). Next, we conducted in vitro transcriptional assays and tested the contribution of the FoxO1–Dio2 axis to THR-dependent reporter gene expression (Supplementary Fig. 8C). We noted that addition of T4 to HEK293 cells, co-transfected with a luciferase vector harboring two TH-responsive elements (TREs) along with expression plasmids for TRα1, the predominant isoform of THR in cardiomyocytes[28,29], and caFoxO1 resulted in significant and concentration-dependent induction of luciferase activity (Supplementary Fig. 8D), supporting the notion that FoxO1-dependent

induction of *Dio2* is required to potentiate TRα1 activity in the presence of T4.

To test this further, we transfected luciferase vector alone or in combination with TRα1, caFoxO1, or both, and assessed luciferase activity. We observed that co-transfection of both FoxO1 and TRα1 was essential for robust and synergistic activation of luciferase activity in the presence of T4, whereas addition of T4 resulted in modest induction of luciferase activity in cells co-transfected with TRα1 alone (Fig. 7c). By contrast, addition of T4 had no effect on basal luciferase activity or in cells co-transfected with FoxO1 alone (Fig. 7c), suggesting exogenous TRα1-dependent induction of reporter activity. Furthermore, increased luciferase activity elicited by FoxO1 may stem from a cryptic FoxO1 binding site within the luciferase vector.

Finally, we analyzed THR-responsive reporter luciferase activity in *FoxO1*- and *Dio2*-deficient NRVMs (Supplementary Fig. 8C). We noted that addition of T4 resulted in robust induction of reporter activity in control siRNA-treated cells (Fig. 7d, e), indicating that endogenous Dio2 activity was sufficient to activate TRα1 transcriptional activity. Consistent with this notion, T4-dependent activation of reporter activity was blunted in both *FoxO1*- (Fig. 7d) and *Dio2*-specific (Fig. 7e) siRNA-treated cells, respectively. However, selective knockdown of *FoxO1* and *Dio2* had only a modest effect on basal reporter activity (Fig. 7d, e). These data suggest that Dio2-dependent increases in intracellular active TH levels (i.e. T4 to T3 conversion) are essential for T4-dependent activation of reporter activity. To probe this notion further, as well as test the physiological relevance of PTU action on Dio2 to attenuate stress-induced cardiac hypertrophy, we evaluated whether PTU can inhibit T4-dependent activation of reporter activity and whether that inhibitory effect of PTU can be rescued by T3 (Supplementary Fig. 8C). Indeed, we noted concentration-dependent inhibition of TH-responsive reporter activity by PTU, whereas addition of T3 along with the highest dose of PTU resulted in complete rescue of reporter activity (Supplementary Fig. 8E). Taken together, our findings support a model in which FoxO1-dependent reciprocal regulation of *Dio2* and *Dio3* modulates intracellular TH metabolism, contributing importantly to stress-dependent adaptive or maladaptive hypertrophic growth of cardiomyocytes (Fig. 7f).

## Discussion

This study unveils a FoxO1–Dio2 signaling axis as a previously unrecognized mechanism of stress-induced adaptive and maladaptive remodeling of cardiomyocytes (Fig. 7f). Specifically, we report five important findings. First, we report that FoxO1, but not FoxO3, activity is essential for reciprocal regulation of *Dio2* and *Dio3* gene expression in cardiomyocytes. Second, utilizing FoxO1 loss- and gain-of-function strategies, we have identified *Dio2* as a direct downstream target of FoxO1 and observe that FoxO1 binding to a conserved FRE within the *Dio2* promoter is required to activate *Dio2* expression. Third, we have discovered a previously unrecognized FoxO1–Dio2 axis as a critical driver of TH-induced cardiomyocyte growth. Fourth, we have discovered activation of FoxO1 in early post-TAC heart, a finding corroborated by analysis of global gene expression patterns[55]. Moreover, load-induced expression of FoxO1 targets was blunted in cKO mice, pointing to FoxO1 specificity. Fifth, we have uncovered an essential role for TAC-induced activation of FoxO1 in pathological cardiac remodeling, with afterload-induced cardiomyocyte growth and remodeling significantly blunted in mice in which FoxO1 was silenced selectively in cardiomyocytes. Furthermore, we report that both pharmacological inactivation of Dio2 activity or depletion of Dio2 levels attenuated cardiomyocyte hypertrophic growth and TH-responsive gene expression. Finally and

from a larger perspective, our findings highlight the importance of intracellular TH homeostasis and metabolism in the cellular response to disease-related stress.

Disease-related stress elicits a maladaptive myocardial remodeling response that is associated with activation of a fetal gene program, fibrosis, contractile dysfunction, and cardiomyocyte hypertrophy, which together culminate ultimately in heart failure, a huge burden on individuals and society[1]. In order to contemplate suppression of pathological cardiac remodeling as a therapeutic intervention, identification and elucidation of proximal triggers are required. Given the established roles of FoxO transcription factors in a wide array of cellular functions and organ homeostasis[12], we set out to decipher the role and molecular mechanisms of FoxO1 action in pathological cardiac remodeling in the setting of the clinically prevalent circumstance of elevated afterload (e.g. hypertension).

FoxO1 phosphorylation by Akt and deacetylation by Sirt1 are the most widely studied mechanisms of FoxO1 inactivation[24] and activation[56], respectively. Recently reported studies, however, have uncovered significant cooperativity between TH and Sirt1 in FoxO1 activation by inhibiting Akt function[57]. As such, data reported here support a functional link between FoxO1 and TH in gene expression and cellular growth. Several lines of evidence support the notion that early activation of FoxO1 in load-stressed heart is a major driver of TAC-induced pathological cardiac remodeling, including decreased FoxO1 phosphorylation and attenuation of TAC-induced induction of select FoxO1 target expression in cKO TAC heart. Furthermore, genetic inactivation of *FoxO1* in cardiomyocytes blunted TAC-induced cardiomyocyte hypertrophic growth, reactivation of the fetal gene program, elicitation of cardiac fibrosis and contractile dysfunction. Although the precise mechanism of FoxO1 activation in early post-TAC heart remains unclear, it is plausible that cooperative actions of TH and Sirt1 contribute by suppressing Akt activity. This notion is supported by the observation that mice lacking Akt1 are sensitized to TAC-induced hypertrophy[58], whereas mice lacking Sirt1 manifest resistance to cardiac hypertrophy[59]. However, we cannot formally rule out a possible role of yet unidentified factor(s) in this phenomenon.

Reciprocal regulation of *Dio2* and *Dio3* expression in skeletal muscle cells by FoxO3 has been reported to govern skeletal muscle cell growth and regeneration[33,60]. Utilizing FoxO loss- and gain-of-function strategies, we have uncovered that FoxO1, but not FoxO3, is primarily involved in reciprocal regulation of *Dio2* and *Dio3* expression in stressed cardiomyocytes. Importantly, FoxO1 activity had no significant effects on *Dio1* expression in cardiomyocytes. Moreover, several lines of evidence support the notion that FoxO1 is a direct, upstream regulator of the *Dio2* gene in cardiomyocytes. Collectively, our data have identified a previously unrecognized FoxO1–Dio2 axis critically involved in stress-induced adaptive and maladaptive growth of cardiomyocytes. Thus, our study and previously reported work in skeletal muscle[33,60] highlight a specific and essential role for FoxO1 and FoxO3 in intracellular TH homeostasis in cardiac and skeletal muscle cells, respectively. Presently, it is unknown whether FoxO factor-mediated reciprocal regulation of *Dio2* and *Dio3* expression in distinct muscle cell types is context dependent or responsive to yet unidentified cell-specific co-factor(s).

Pathological stress-induced cardiac hypertrophy is known to be governed by the intricate and highly orchestrated actions of numerous transcriptional, signaling, and metabolic events and epigenetic modifications of chromatin structure[9]. In addition, TH signaling and TRα1 activity are intimately linked to both physiological and pathological cardiac growth[28,29]. In heart failure patients, circulating TH levels are often low[28,29], whereas *Dio2* expression increases in pathologically hypertrophied heart[49]. On the other hand, gain or loss of TRα1 activity can potentiate and

prevent pathological cardiac hypertrophy, respectively[61], suggesting that intracellular TH status is an important and independent determinant of cardiomyocyte health. Although it has been reported that elevated T3 levels improve contractile activity of hypertrophic cardiomyocytes[54], other studies demonstrated that T3 treatment had no[62], or detrimental[63,64], effects on contractile activity in hypertrophied heart. Together, these studies suggest that the beneficial and detrimental effects of TH depend on the nature of the stress and the extent and duration of exposure, thereby underscoring the complexities of TH actions in cardiac remodeling. Of note, treating heart failure patients with TH has been limited due, in part, to cardiotoxic actions[64].

Several lines of evidence, including attenuation of T4-induced NRVM growth in Dio2-, but not Dio1-, deficient cells and robust induction of Dio2, but not Dio1, expression in TAC-stressed hearts and cardiomyocytes, suggest strongly that Dio2 is the primary deiodinase enzyme involved in intracellular TH metabolism and homeostasis. Moreover, attenuation of TH-responsive reporter activity in PTU-treated and in Dio2-deficient cardiomyocytes, as well as TAC-induced cardiac hypertrophy in PTU chow-fed WT mice, collectively provide additional credence to the notion that PTU-dependent inhibition of Dio2 activity is intimately linked to attenuation of TAC-induced cardiomyocyte growth. These results are consistent with a prior study[49] demonstrating that PTU treatment significantly blunted oxidative stress-induced cardiac damage and remodeling. Importantly, oxidative stress resulted in robust elevation of Dio2, but not Dio1, expression and cardiac T3 levels in hypertrophic heart, which were blocked by PTU treatment[49].

Several studies, however, have reported that the deiodination activity of Dio2 in tissue homogenates is insensitive to PTU or requires high concentrations of PTU for efficacy[65–67]. This discrepancy could be related to the fact that an unidentified cellular reducing factor is critical for efficient Dio2 deiodination activity in vivo[65]. To mimic in vivo conditions, most in vitro studies therefore incorporate high concentrations of the reducing chemical, dithiothreitol (DTT), to activate Dio2 activity within the homogenate. Of note is the fact that PTU is known to compete with DTT[65]. Moreover, subtle differences between Dio1 and Dio2 in the amino acid sequences around the enzymatic active site have profound effects on Dio2's affinity for DTT as well as its PTU sensitivity[65]. For example, Dio1 activity of Tilapia is insensitive to PTU in vitro[65,68]. Subsequently, it was reported that the PTU-insensitive nature of Tilapia Dio1 is not due to a serine-to-proline substitution at position 128 within the enzymatic active site (as compared with other mammalian Dio1 enzymes), but rather stems from differences in amino acid sequence elsewhere within the protein[68]. Mammalian Dio2 also harbors a proline at that position[65]. Therefore, it will be important to determine whether the unidentified cellular reducing factor and DTT activate Dio2 similarly and whether PTU differentially affects Dio2 activity in vitro and in vivo. Future work will be essential to assess PTU-dependent inhibition of Dio2 activity without DTT[49].

We have uncovered a specific and previously unrecognized FoxO1–TH signaling cascade that is critically involved in hypertrophic and pathological remodeling of the heart. We have identified Dio2 as a direct downstream target of FoxO1 and unveiled the circuitry whereby FoxO1-dependent reciprocal regulation of Dio2 and Dio3 expression is a critical driver in TH-induced cardiomyocyte growth. Further, our findings highlight the importance of intracellular TH homeostasis in the cellular response to disease-related stress. These findings have potential therapeutic relevance, as we[69–71] and others[72] have reported that both genetic and pharmacological inhibition of histone deacetylase activity blunts, and can even reverse, stress-induced cardiac remodeling. Given that acetylation and deacetylation modulate

FoxO1 transcriptional activity and deiodinase gene expression[24], our findings point to potential translational strategies involving small molecules in current clinical use.

## Methods

**Animals, echocardiography, and TAC.** To generate cardiomyocyte-specific FoxO1 knockout (cKO) animals, mice harboring a floxed FoxO1 allele were crossed with αMHC-Cre transgenic mice. FoxO1-cKO and corresponding control floxed or αMHC-Cre mice were maintained on an FVB genetic background. The cKO mice were fertile and developed normally with no apparent cardiovascular phenotype. Transthoracic echocardiography was performed on conscious, gently restrained mice using Sonos 5500 and VisualSonic Vevo 2100 systems[73]. Briefly, LVEDD and LVESD were measured from M-mode traces recorded at the level of the papillary muscles before and 3 weeks after surgery. Fractional shortening (FS) was calculated as (LVEDD − LVESD)/LVEDD and expressed as a percentage. Male mice (8–10 weeks old) of respective genotypes with normal FS were subjected to pressure overload by TAC along with a cohort exposed to sham (control) surgery[51]. At defined time points, animals were euthanized and whole heart or LV were collected for histological, molecular, and biochemical analyses. All mice were maintained according to the guidelines of the Care and Use of Laboratory Animals published by the US National Institutes of Health and the Institutional Animal Care and Use Committee, and all procedures were approved by the Institutional Ethics Review Committee of the University of Texas Southwestern Medical Center.

**Neonatal cardiomyocyte isolation, transfection and adenovirus infection.** Primary NRVM were isolated and cultured in Dulbecco's modified Eagle's medium (DMEM)/M199 (3:1) containing 3% fetal bovine serum (FBS) and antibiotics[74]. For gene knockdown, NRVM were transfected with two sequence-independent, gene-specific siRNAs (Sigma) using Lipofectamine RNAiMax (Invitrogen) in Opti-MEM (Gibco). After 6 h, cells were washed twice with growth medium and cultured overnight in growth medium. For adenovirus-mediated protein over-expression, cells were incubated for 1 h with adenovirus at a multiplicity of infection of 50 plaque-forming units per cell. Cells were then washed twice with fresh serum-free DMEM/M199 medium and cultured for an additional 24 h. Under these conditions, infection efficiency routinely exceeded 90%.

**RNA and genomic DNA isolation for semi- and quantitative reverse transcription PCR.** Total RNA (0.3–1 μg) isolated from neonatal mouse ventricles, adult LV, liver, adult ventricular cardiomyocytes, and NRVM was used to prepare cDNA using the iScript cDNA synthesis kit (Bio-Rad)[74]. Real time PCR and semi-quantitative PCR was performed using 5- to 10-fold diluted cDNA, Taqman or gene-specific primers and SYBR Green on a Roche light cycler 480. Primer sequences are provided in Supplementary Table 1. Ribosomal 18S RNA and actin were used as control for quantitative and semi-quantitative analyses, respectively. Quantitation of mRNA levels of specific genes was analyzed using a ΔΔCt method.

Genomic DNA (gDNA) from neonatal mouse ventricle, tail, and adult ventricular cardiomyocytes was isolated using Quick-DNA miniprep plus kit according to the manufacturer's instructions (Zymo Research). Semi-quantitative PCR analyses using equal amounts of gDNA (10–100 ng) from WT and cKO mice were used to assess Cre-mediated recombination of loxP sites in the FoxO1 gene.

**PTU treatment.** TH deficiency was induced by feeding animals for the indicated durations with iodine-free chow supplemented with 0.15% PTU purchased from Harlan Teklad Co. (TD 97061) (Madison, WI). Mice, fed normal or PTU chow for a week, were subjected to sham or TAC surgery and followed for 3 weeks. Hearts and/or LVs were harvested to evaluate cardiac hypertrophy, fibrosis, and gene expression.

**Immunoblotting.** Whole-cell lysates of mouse LV and NRVMs were prepared in T-PER (Thermo) containing protease and phosphatase inhibitors (Roche). Nuclear fractions of heart 4 days following sham- or TAC procedures were prepared using NE-PER nuclear cytoplasmic extraction reagents according to the manufacturer's instructions (Thermo). Equal amounts of protein (3–10 μg) were separated on a 4–20% SDS-PAGE gel (Bio-Rad), transferred to a nitrocellulose membrane, and immunoblotted to detect and quantify specific protein bands using an Odyssey scanner (LI-COR version 3)[74]. Proteins were detected with a 1000-fold dilution of the following primary antibodies: FoxO1 (#2880, Cell Signaling and ab39670; Abcam); Phospho-FoxO1 (#9464, Cell Signaling), ERK (#4695, Cell Signaling); Phospho-ERK (#4370, Cell Signaling); 10,000-fold dilution of GAPDH (10R-G109a, Fitzgerald) and tubulin (ab6046; Abcam) and 500-dilution of Dio2 antibody (ab77481, Abcam) antibodies, respectively.

**[3H]Leucine incorporation.** Leucine incorporation was determined in NRVMs transfected with control or gene-specific siRNAs. After transfection in Opti-MEM, cells were cultured in serum-free medium for 24 h, washed three times with serum-free medium, and then treated with either vehicle or T4 or T3 (100 nM), Ang II (200 nM), IGF-1 (50 nM), and PE (50 μM)[70]. In some assays, NRVMs were transfected with control or gene-specific siRNA and TH-induced NRVM growth

was assessed in the presence of DMSO (vehicle) or PTU (0.5 mM). Relative growth of control siRNA-transfected and vehicle-treated NRVMs was set to 100%.

**Histology and immunohistochemistry**. Histology and immunohistochemical (IHC) analyses were performed with formaldehyde-fixed hearts and processed as routine paraffin sections[74]. Briefly, hearts were fixed in 4% paraformaldehyde overnight at 4 °C and transferred to 1× PBS, followed by paraffin embedding. After deparaffinization and antigen retrieval, hematoxylin/eosin staining was performed for morphological analysis. Wheat germ agglutinin was used for CSA measurements, quantified from at least 30 cells per section and three independent heart sections per group. Masson trichrome staining was used for measurements of fibrosis. IHC analyses of phospho-histone 3 and cardiac troponin T were performed to detect proliferating cardiomyocytes[35] and DAPI was used to stain nuclei.

**Luciferase and ChIP assay**. Transcriptional assays using pDR4, a luciferase vector harboring two TREs with or without constitutively active FoxO1 (caFoxO1) and TR1α expression vectors, were performed in HEK293 (Clontech; cat# 632271) and NRVM. Plasmids were transfected into HEK293 cells using Fugene HD (Roche)[75]. On the other hand, pDR4 luciferase vector was transfected into NRVM using Lipofectamine 2000 (Invitrogen). Luciferase assays using a 700-bp Dio2 reporter construct harboring WT or mutated FoxO-response elements (FRE) were performed in COS7 cells (ATCC; CRL-1651) with or without caFoxO1. Luciferase activity with or without TH and PTU treatment was measured using dual luciferase kit (Promega) and normalized with renilla activity[16]. All cell lines tested negative for mycoplasma contamination.

Quantitative analyses of ChIP assays were conducted to assess FoxO1 occupancy at the Dio2 promoter in ventricular tissues of Sham and TAC-operated mice (4 days) using the Zymo-Spin ChIP kit according to the manufacturer's instructions (Epigenetics)[74]. Briefly, isolated ventricular tissues were minced and exposed to 1% formaldehyde with gentle shaking (15 min, RT). After quenching the crosslinking reaction with glycine, tissue was washed twice with PBS, and then homogenized in nuclear extraction buffer. The homogenate was centrifuged (10 min, 500g, 4 °C) and the supernatant was discarded. The nuclear pellet was resuspended in nuclear lysis buffer, sonicated, and centrifuged (15 min, 12,000g, 4 °C) to prepare chromatin solution. Chromatin solution was diluted 10-fold and incubated with control (IgG) and FoxO1 sera (3 μg) overnight at 4 °C with rotation. DNA purified from chromatin solutions was used as input. Purified DNA with or without immunoprecipitation was used to determine relative FoxO1 occupancy by comparing the Ct value of 30-fold diluted input DNA with undiluted immunoprecipitated DNA samples in control (IgG) and FoxO1 sera.

**Statistical analysis**. Data are presented as mean ± S.E.M. of multiple independent replicates. Data were analyzed using GraphPad Prism software 8.0 and statistical analyses were conducted by the two-tailed, unpaired Student's t-test for experiments with two groups. Values of $p \leq 0.05$ were considered statistically significant. All experiments were performed with at least three biological replicates. No statistical analysis was performed to predetermine sample sizes. Mice were randomly assigned to experimental and control groups. Investigators were blinded to the genotypes of animal during the experiments and outcome assessments.

**Reporting summary**. Further information on research design is available in the Nature Research Reporting Summary linked to this article.

## Data availability

The authors declare that the data supporting the findings of this study are available within the paper and its Supplementary Information. Each data point corresponding to figures that describe the results from in vivo and/or in vitro model studies are provided as separate Source Data for Figs. 1a–f, 2a, b, d–f, 3a–f, 4b, 5b–d, 6a–f and 7a–e and Supplementary Figs. 1A–D, 2A–C, 3A–E, 4B–F, 5A–F, 6C, 7A, C and 8A, B, D, E. All primary and Supplementary data will be provided by the corresponding author upon reasonable request.

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

## Acknowledgements

We thank the entire Hill lab for constructive discussions. We thank the Histology Core and Diana C. Canseco for help with immunohistochemistry and Dr. Yauk of Carleton University, Ontario, Canada, for reagents. The work was supported by grants from the NIH (HL-120732; HL-128215; HL-126012; HL-147933 to J.A.H.; HD-101006 to B.A.R; HD-087351 to B.A.R.), American Heart Association (14SFRN20740000 to J.A.H.; 19TPA34920001 to B.A.R.; 19CDA34680003 to F.A.; 18POST34060230 to G.G.S.), CPRIT (RP110486P3 to J.A.H.), the Leducq Foundation (11CVD04 to J.A.H.), the Comision Nacional de Investigacion Cientifica y Tecnologica de Chile Fondo de Financiamiento de Centros de Investigacion en Areas Prioritarias (FONDAP; grant 15130011 to S.L.), and Fondo Nacional de Desarrollo Científico y Tecnologico (FONDECYT grant1161156 to S.L.); ZVW was supported by a Scientist Development Grant (SDG) from the AHA (14SDG18440002). D.L.L. was supported by a predoctoral fellowship from the AHA (14PRE19770000).

## Author contributions

A.F. and J.A.H. designed research; A.F., Z.V.W., Y.L., D.L.L., X.L., G.G.S., F.A., H.I.M., P.K.B., and A.N. performed research; B.A.R. and S.L. contributed new reagents/analytical tools; A.F. and T.G.G. analyzed data; and A.F. and J.A.H. wrote the paper and B.A.R., S.L., and T.G.G. edited it.

## Competing interests

The authors declare no competing interests.
