## [Peer Review File · Nature Communications]

Reviewers' comments:

Reviewer #1 (Remarks to the Author):

My comments from the original submission have been addressed. The manuscript in its current form is much improved with revisions and additional data.

Reviewer #2 (Remarks to the Author):

In this paper, Ferdous et al investigate on the role of FoxO family members (particularly FoxO1) in stress-induced remodeling of cardiomyocytes. They found that

FoxO1, but not FoxO3, activity is essential for reciprocal regulation of type II and III iodothyronine deiodinases key enzymes involved in intracellular TH metabolism.

The topic of the paper is interesting, although the paper is somehow not as novel as the authors claim. Many studies already investigated on the role of foxO1 in cardiac remodeling and a similar FoxO3 –Dio2 axis has already been described in skeletal muscle.

Finally, but highly relevant, authors used PTU as a selective and specific inhibitor of D2 enzymatic activity in their in vitro and in vivo studies. This is not correct, since D2 enzyme activity has been characterized as PTU-resistant, () so most of the conclusions in the paper are affected by this inappropriate tool.

Specific comments

While there are several reports that acute PTU treatment in vivo blocks type 1 but not type 2 activity (Silva JCI 1982, Silva JCI 1985, Bianco AC. Endocrine Rev 2002), the articles cited by the authors in support of an inhibitory effect of PTU on D2 are based on indirect evidence. In any case, since the PTU concentrations used in this and in the cited articles differ widely, it is not inconceivable that PTU inhibited also D2 in experimental conditions used.

To demonstrate this concept, in the absence of genetically-depleted D2 mice, the authors should:

1) Demonstrate that D2 is effectively blocked in vivo in their experimental settings by PTU in other D2-expressing tissues.

For example, the authors could analyze brown adipose tissue and evaluate whether, in PTU-treated animals, the thermogenesis is impaired in cold stimulated animals (De Jesus JCI 2001) or that the UCP-1 mRNA response to NE is blunted in PTU-treated animals (De Jesus JCI 2001).

2) Demonstrate that in their experimental setting PTU does not affect plasma T3 and T4 levels in normal and in early post-TAC hearts conditions. This would indeed prove that the PTU-effect is "deiodinase"-mediated and not due to altered circulating thyroid hormone levels.

Reviewer #3 (Remarks to the Author):

In my opinion, the authors satisfactorily addressed the concerns previously raised by former

Reviewer 3. I have no further comments.

We are pleased that all three reviewers found our responses to most of the comments and critiques satisfactory. Importantly, Reviewer #1 and the independent Reviewer #3 have no further comments/concerns and found our revised manuscript to be much improved. Similarly, Reviewer #2 also found our responses satisfactory to the majority of the comments and critiques voiced from our original submission, yet raised issues regarding (i) priority and (ii) technical aspects. In response, we have conducted additional experiments and added the resulting findings to the end of our responses here and incorporated the data in a revised Online Supplemental Data section of the revised manuscript.

Here, we provide a detailed point-by-point response to each of the reviewers' comments.

Response to Reviewer #1:

My comments from the original submission have been addressed. The manuscript in its current form is much improved with revisions and additional data.

Response: We thank the Reviewer for these positive comments.

Response to Reviewer #2:

General comments:

In this paper, Ferdous et al investigate on the role of FoxO family members (particularly FoxO1) in stress-induced remodeling of cardiomyocytes. They found that FoxO1, but not FoxO3, activity is essential for reciprocal regulation of type II and III iodothyronine deiodinases key enzymes involved in intracellular TH metabolism.

The topic of the paper is interesting, although the paper is somehow not as novel as the authors claim. Many studies already investigated on the role of foxO1 in cardiac remodeling and a similar FoxO3 –Dio2 axis has already been described in skeletal muscle.

Finally, but highly relevant, authors used PTU as a selective and specific inhibitor of D2 enzymatic activity in their *in vitro* and *in vivo* studies. This is not correct, since D2 enzyme activity has been characterized as PTU-resistant, () so most of the conclusions in the paper are affected by this inappropriate tool.

Response: We thank the Reviewer for these comments. In our revised manuscript, we have comprehensively addressed your concerns regarding both (i) priority and (ii) technical issues by providing substantial new *in vitro* and *in vivo* data along with discussion of numerous previously reported studies to support our overall hypothesis that the FoxO1-Dio2 axis plays an essential role in stress-induced hypertrophic growth of cardiomyocytes.

(i) Priority: In the current study, we are studying FoxO1 and cardiac myocytes, defining a unique mechanism that involves intracellular thyroid hormone metabolism. An entirely different transcription factor with distinct, independent activity, FoxO3, does something analogous in another very different type of muscle cell, skeletal myocytes. It is important to note that in that study, loss- and/or gain-of-function strategies of other FoxO (e.g. FoxO1) and deiodinase (e.g. Dio1 and Dio3) genes were NOT utilized to probe the specific role of FoxO3-Dio2 axis in skeletal myocytes. By contrast, we utilized a combination of loss- and gain-of-function strategies of a specific gene and a series of biochemical techniques to uncover an essential and specific role of the FoxO1-Dio2 axis in hypertrophic growth of stressed cardiomyocytes *in*

vitro and *in vivo*. Importantly, the role of FoxO1 and Dio2 cannot be compensated by FoxO3 (Figure 1F) or Dio1 (Figures 1B, S2B and S5), respectively, hence this highlights the novelty of our study.

Secondly, whereas we and others have reported the role of FoxO family members (e.g. FoxO1, FoxO3 and FoxO4) in cardiovascular development and stress-induced cardiac remodeling (Ref# 13-20), none of these prior studies addressed the role and underlying mechanism of FoxO1 action in pathological cardiac remodeling during the very clinically important conditions of pressure overload stress.

Finally, in contrast to the FoxO3-Dio2 study in skeletal muscle growth, our study has uncovered a previously unrecognized role and mechanism of FoxO1 in maladaptive cardiac hypertrophy in the setting of a clinically prevalent circumstance of elevated afterload (e.g. hypertension), thereby highlighting the clinical relevance of our study and shedding light on a potential therapeutic strategy for the future. Therefore, we respectfully submit that our study is, in fact, novel and will be of general interest.

Specific comments

While there are several reports that acute PTU treatment *in vivo* blocks type 1 but not type 2 activity (Silva JCI 1982, Silva JCI 1985, Bianco AC. *Endocrine Rev* 2002), the articles cited by the authors in support of an inhibitory effect of PTU on D2 are based on indirect evidence. In any case, since the PTU concentrations used in this and in the cited articles differ widely, it is not inconceivable that PTU inhibited also D2 in experimental conditions used.

Response: We hope that the Reviewer will agree with the fact that most of the conclusions of our study were independent of PTU data. However, we agree with the Reviewer's view that a higher PTU concentration may account for its ability to inhibit Dio2 activity. We would also add that in addition to PTU concentration, duration of PTU treatment might be critical to inhibit Dio2 activity *in vivo*. For example, long-term (1-16 weeks) treatment with high concentrations of PTU resulted in marked reduction of cardiac T3 [Wang YY et al., *Cardiovascular Res.*, 2010 (Ref # 49)] as well as serum T4 and T3 levels in mice, rat, horse, and human, respectively (Ref #49; Cooper DS., *Endocrinology*, 1983; Johnson PJ, *Equine Vet. J.*, 2003; Geffner DL., *JCI*, 1975) with concomitant increase in serum and thyroid PTU concentration (Cooper DS, 1983). By contrast, in the indicated studies (e.g. Silva JE, *JCI* 1985), rats were treated (IP) twice with PTU (3mg/100g BW) only 30 minutes prior to and 17 hours after injecting radiolabeled T4 in jugular vein and analyzed radiolabeled serum T4 and T3 levels and deiodinase activity after 0, 3 and 6 hours of second PTU injection. A substantial fraction of IP-injected drugs are known to accumulate initially in the liver, and the IC₅₀ of PTU to inhibit Dio1 activity *in vitro* is extremely low (1.3µM) (Renko K, *Endocrinology*, 2012). Therefore, it is conceivable that short-term PTU treatment was sufficient to inhibit liver Dio1 activity but might not be sufficient to reach effective PTU concentrations in other tissues, including brown adipose tissue, to affect Dio2 activity. As such, short-term PTU treatment manifested only a modest effect on kinetics of radiolabeled serum T4 and T3 levels (Silva JE, *JCI* 1985).

Collectively, these studies suggest strongly that marked reduction of both tissue and serum TH levels most likely resulted from PTU-dependent inhibition of both Dio1 and Dio2 activity since mice with global knockout of either Dio1 (Schneider MJ, *Endocrinology*, 2006) or Dio2 (Schneider MJ, *Mol. Endocrinology*, 2001) had no significant effect on serum T3 levels. Accordingly, it is plausible that inhibition of Dio2 activity following long-term PTU treatment manifested protective effects on stress-induced maladaptive cardiac remodeling and fibrosis

(Ref #49) and high fat diet (HFD)-induced metabolic derangement and obesity in mice [Vernia S., et al, *Genes & Dev.*, 2013 (Ref #45)].

Similarly, we have also demonstrated that compared with control chow-fed mice, TAC-induced maladaptive cardiac hypertrophy and fibrosis were significantly attenuated in 4-week PTU chow-fed mice (Fig. S4). To test for a direct effect on cardiomyocytes, we utilized knockdown of specific genes in NRVM and recapitulated the *in vivo* data by demonstrating that the FoxO1-Dio2 axis is essential for thyroid hormone (TH)-induced hypertrophic growth of cardiomyocytes (Figs. 1A, 1B, 2B), which cannot be compensated by FoxO3 (Figure 1F) or Dio1 (Figures 1B, S2B and S5). Importantly, we have also demonstrated that T4-induced NRVM growth was still sensitive to PTU (500 μ M) in the absence of *Dio1*, but no longer sensitive in *Dio2*-deficient NRVMs (Figures S5E and S5F). This bears repeating; the PTU-induced inhibition of hypertrophic growth we observe in cardiomyocytes is not correlated with the presence of Dio1 but instead with Dio2. This suggests that under these conditions either PTU can inhibit Dio2 or inhibits some unknown (not Dio1) protein that then acts through Dio2 to result in the phenotype. Occom's razor suggests the former.

Although our data, along with previously reported work, suggest strongly that PTU can inhibit Dio2 activity *in vitro* and *in vivo* at certain dose and experimental settings, we undertook additional experiments to test this possibility directly, as suggested by the Reviewer.

Specific comment #1: Demonstrate that D2 is effectively blocked *in vivo* in their experimental settings by PTU in other D2-expressing tissues.

For example, the authors could analyze brown adipose tissue and evaluate whether, in PTU-treated animals, the thermogenesis is impaired in cold stimulated animals (De Jesus JCI 2001) or that the UCP-1 mRNA response to NE is blunted in PTU-treated animals (De Jesus JCI 2001).

Response: We agree with the Reviewer that analyses of PTU effect on thermogenesis in brown adipose tissue (BAT) of mice following cold exposure or norepinephrine (NE)-dependent induction of *UCP1* expression in BAT (De Jesus JCI 2001) would reveal a specific effect of PTU on Dio2 activity. However, we respectfully submit that analysis of such technically complex biology is beyond the scope of our expertise and focus on cardiovascular biology. Nevertheless, we have tested PTU effects on NE-dependent induction of *UCP1* gene expression in adipocytes *in vitro*. BAT stromal vascular fraction cells (SVC) (kindly provided by Dr. Rana K. Gupta of UT Southwestern Medical Center) were cultured and differentiated to adipocytes (panel A) as described previously (Shao M., et al., *Cell Metabolism*, 2016), except that 10nM T4 was added in differentiation medium. Consistent with the reported study (Shao et al., 2016), robust induction of Dio2, but not Dio1, gene expression (panel B) and detection of *UCP1* transcripts (panel C) in adipocytes indicates efficient differentiation of SVC into adipocytes. Importantly, compared with untreated adipocytes, marked induction of *UCP1* mRNA levels in NE-treated cells was significantly attenuated in the presence of PTU (panel C). These data again suggest strongly that PTU can inhibit Dio2 activity *in vivo* under certain dose and experimental setting.

In light of these data and a previously reported study (De Jesus JCI 2001), we further undertook analogous experiments in NRVM. We have demonstrated that specific knockdown of *FoxO1* and *Dio2* in NRVM significantly attenuates T4-induced, but not basal, thyroid hormone (TH)-responsive reporter activity (Figs. 7C, 7D). We therefore tested (1) whether PTU can inhibit the T4-induced TH-responsive reporter activity, and (2) whether that inhibitory effect can be rescued by exogenous T3. Indeed, we noted concentration-dependent inhibition of TH-responsive

reporter activity by PTU (panel E). Importantly, addition of T3 in the presence of the highest dose of PTU resulted in complete rescue of reporter activity (panel E). These data not only support the notion that PTU inhibits Dio2 activity *in vivo* at higher doses but additionally support the physiological relevance of PTU action on Dio2 to attenuate stress-induced cardiac hypertrophy (Figs. S4; S5F; Ref #49).

We have incorporated these important NRVM data in the Online Supplemental (please see Fig. S8E) of the revised manuscript and revised text in Results (page 14) and Discussion (page 17) sections and in Figure legend accordingly.

Specific comment #2: Demonstrate that in their experimental setting PTU does not affect plasma T3 and T4 levels in normal and in early post-TAC hearts conditions. This would indeed prove that the PTU-effect is “deiodinase”-mediated and not due to altered circulating thyroid hormone levels.

Response: Consistent with a previously reported study (Ref #49), we have demonstrated that compared with control chow-fed mice, TAC-induced maladaptive cardiac hypertrophy and fibrosis were significantly attenuated in PTU chow-fed mice (Fig. S4). Compared with normal chow-fed mice, we also noted robust induction of β MHC and attenuation of α MHC expression in LVs of PTU chow-fed mice, indicating blockade of T3 biosynthesis (Ref #43). Based on the reported effect of long-term PTU treatment on reduced serum TH levels in animal and human (Ref #49; Cooper DS., *Endocrinology*, 1983; Johnson PJ, *Equine Vet. J.*, 2003; Geffner DL., *JCI*, 1975), it is conceivable that compared with control chow-fed mice, long-term (4-weeks) PTU chow-fed mice might have reduced serum TH levels. Therefore, we respectfully submit that analyses of serum TH levels under our experimental setting may not provide meaningful results to address the Reviewer’s concern.

Response to Reviewer #3:

In my opinion, the authors satisfactorily addressed the concerns previously raised by former Reviewer 3. I have no further comments.

Response: We thank both current and previous Reviewer #3 for their positive comments.

In the amended manuscript, we have revised the text in response to all the Reviewers’ comments and as the new data warranted. These new data further support our overall conclusion that the novel FoxO1-Dio2 axis plays an important role in cardiomyocyte growth by regulating intracellular TH homeostasis. We have also provided strong evidence that PTU can inhibit Dio2 activity *in vivo* at higher concentrations, and that Dio2 activity is required for intracellular TH metabolism, stress-induced cardiomyocyte growth and TH-responsive gene expression. These data not only support our conclusion but also suggest strongly that transcriptional regulation of Dio2 gene expression by FoxO1 is physiologically relevant.

Figure legend. Inhibition of *Dio2* activity by PTU attenuates thyroid hormone-responsive gene expression in adipocytes and cardiomyocytes. (A) Schematic of brown adipose tissue-stromal vascular fraction cells (BAT-SVC) differentiation to adipocytes, treatment with or without norepinephrine (NE) and propylthiouracil (PTU) and quantitative RT-PCR (qPCR) analyses of mRNA levels of specific gene. (B) qPCR analyses of mRNA levels of the indicated genes in SVC and adipocytes. Note that compared with SVC, *Dio2* mRNA levels were robustly increased in adipocytes, while mRNA of *Dio1* in both SVC and adipocytes was essentially undetectable. Relative *Dio2* mRNA levels in SVC was set to 1. (C, D) qPCR analyses of mRNA levels of *UCP1* (C) and the indicated genes (D) in adipocytes with or without NE and PTU treatment. Note that addition of PTU significantly attenuated *UCP1* expression only in NE-treated adipocytes (C), and that inhibition was not associated with dysregulation of *Dio2* or *Dio1* expression (D). Relative *UCP1* (C) and *Dio2* (D) expression in control PBS (cont)-treated cells was set to 1. (E) T4-dependent activation of TH-responsive reporter activity in NRVM reveals concentration-dependent marked attenuation of reporter activity by PTU (n = 3). Luciferase activity in vehicle (Veh)-treated cells was set to 1. Note that addition of T3 (10nM) completely rescued the inhibitory effect of PTU. In each panel, data are depicted as mean \pm SEM (n = 3). * p <0.05 vs control; ** p <0.01 vs control; # p <0.05 vs control; ## p <0.01 vs control; *** p <0.001 vs control. NS, not statistically significant.

REVIEWERS' COMMENTS:

Reviewer #2 (Remarks to the Author):

All major comments from the initial submission have been addressed. The manuscript in its current form is improved with revisions and additional data.